# An Online Method to Detect Urban Computing Outliers via Higher-Order Singular Value Decomposition

**DOI:** 10.3390/s19204464

**Published:** 2019-10-15

**Authors:** Thiago Souza, Andre L. L. Aquino, Danielo G. Gomes

**Affiliations:** 1Grupo de Redes de Computadores, Engenharia de Software e Sistemas (GREat), Departamento de Engenharia de Teleinformática, Universidade Federal do Ceará (UFC), Fortaleza, Ceará CEP 60020-181, Brazil; danielo@ufc.br; 2Instituto de Computação, Universidade Federal de Alagoas (UFAL), Maceió, Alagoas CEP 57072-900, Brazil; alla@laccan.ufal.br

**Keywords:** outlier detection, online monitoring, multiway analysis, HOSVD, MPCA, smart cities

## Abstract

Here we propose an online method to explore the multiway nature of urban spaces data for outlier detection based on higher-order singular value tensor decomposition. Our proposal has two sequential steps: (i) the offline modeling step, where we model the outliers detection problem as a system; and (ii) the online modeling step, where the projection distance of each data vector is decomposed by a multidimensional method as new data arrives and an outlier statistical index is calculated. We used real data gathered and streamed by urban sensors from three cities in Finland, chosen during a continuous time interval: Helsinki, Tuusula, and Lohja. The results showed greater efficiency for the online method of detection of outliers when compared to the offline approach, in terms of accuracy between a range of 8.5% to 10% gain. We observed that online detection of outliers from real-time monitoring through the sliding window becomes a more adequate approach once it achieves better accuracy.

## 1. Introduction

Since 2007, for the first time in human history, more people live in cities than in rural areas. According to the United Nations, by 2030, the world population is expected to be nearly 60% urban, and by 2050, this proportion will increase to 70%. With populations growing, cities need to face increasing and critical problems of urban environments. In this context, the detection of outliers in smart cities scenarios becomes essential to identify hidden patterns from urban data.

The integration between information and communications technology (ICT), cloud computing, and Internet of Things (IoT) in smart cities has contributed to the consolidation of what we call a smart urban environment. This integration assists in the management of the various services offered by the city, such as transportation systems, education, health, safety, and environmental monitoring. In general, these services require online data collections and transmissions, and outdated data may be useless [1]. Thus, online monitoring can uncover hidden patterns, unknown correlations, and the identification of events. Despite the need to accurately identify relevant characteristics of data sets and extract patterns of that help in making the decisions, the problem is that not all the data are relevant, and the extracted information can be biased, noisy, redundant, and incorrect [2,3]. In general, these data types are known as outliers. Outliers are often defined as the observations that appear to be inconsistent with the rest of the dataset, and it is essential to identify them to explore their possible abnormal patterns [4].

In recent years, due to the wide consolidation of smart cities, outlier detection has received increasing research efforts, such as the works geared towards identifying patterns of unusual events in urban traffic flow, trends in air quality change, or water quality monitoring [5,6,7]. The outlier detection task is often performed manually with the help of data visualization tools [8]. This configuration is marked by an offline detection process and on a sample of data in which each user identifies normal patterns, then single out the samples that deviate from the normal patterns [9]. However, the online detection of outliers in real-time intelligent city data is more appropriate compared to offline detection, since online detection must be able to detect deviations at the current point in time in order to notify and/or take actions [8].

A wide range of outliers detection applications can be found in the recent literature [4,10,11,12,13,14], highlighting network security areas [15], cybercrimes [16] and industry [17]. Several other examples of detection of online outliers can be observed in the recent work of [11], where it was revealed that the existing approaches to detect outliers are not effective enough, particularly in detecting them online. In this sense, online outlier detection in urban monitoring data of the smart city is relevant since it can help to monitor physical and atmospheric conditions from an online approach, such as temperature, light, humidity, traffic and other pressures.

In particular, from an environmental perspective, outlier detection from environmental monitoring gains strength in the literature since the detection of noise in cities, water and air pollution, forest conditions, and so on, can provide for a sustainable and intelligent development for cities [10]. In a previous paper, we proposed an offline method to explore the data of a multiway nature of urban spaces in the detection of outliers [18]. This method executes through three stages: in the first one, the data is modeled as a third order data tensor to achieve the reduction of dimensionality in order to obtain a better fit, the second stage is comprised of a classification step, and finally the third step generates a model of identification of refined urban space standards.

Here we propose an online for an outlier detection method that combines multivariate and multidimensional approaches to characterize the behavior patterns of urban environmental monitoring data from high-dimensional structured datasets. We extend our previous method [18] to an online detection approach which: i. uses a sliding window which keeps track of the most recent data and all decomposition and detection tasks performed based on what is "visible" through the window; and ii. evaluates a performance of online outlier detection from both accuracy and receiver operational characteristic (ROC) curves. The use of these steps (i. and ii.) is also a differentiation over traditional offline proposals present in the literature [11]. To evaluate the proposed method, we used real data from environmental monitoring collected from a platform called Smart Citizen [19] of the cities of Helsinki, Tuusula and Lohja. We considered outdoor sensors collecting data for a period of 16 days (December 1 through December 16, 2018). Moreover, because missing values occur frequently throughout the monitoring platform on their various sensor nodes, we chose cities with sensors that had the fewest faults over a continuous period of monitoring. Using these data, it may be possible to detect events as they occur.

The results showed greater efficiency for the online method of detection of outliers when compared to the offline approach in terms of accuracy between a range of 8.5% and 10% gain. Moreover, with the sliding window combined with the tensor factorization, we observed both an incremental stage where the effect of the most recent data is added, while in the decremental stage the effect of the older data is omitted. We observed that online detection of outliers from real-time monitoring through the sliding window becomes a more adequate approach once it achieves better accuracy.

The main contributions of our work are: i. proving an online outlier detection method that combines multivariate and multidimensional approaches providing useful information for improving the planning and operation of cities; ii. combining the multidimensional approach with the sliding window method; iii. generating a dynamic outlier detection threshold as the sliding window for a stream of data where old data expire and new data come in; and iv. evaluating the performance of the proposed method based on real-life datasets. The results showed that our method of online outlier detection is consistently more efficient.

The remaining of the paper is organized as follows: Section 2 shows the description of methodological procedures; Section 3 discusses the experimental results; and Section 4 concludes the work and details some future work.

## 2. Material and Methods

When considering only the multivariate nature in detection (matrix based methods), the outliers may remain invisible. Therefore, we concentrated on a multi-way, which allows us to summarize the high-dimensional data into tensors [20]. For a better understanding of the dynamics of sensed environmental variables, we applied a tensor decomposition method to decompose the environmental big data into relevant patterns, from which we can extract key information related to the semantics of the collected variables. Based on this information, the multivariate approach was then applied to the time series in which we can further reveal the dynamics of cities’ urban environments.

In this section, we introduce our online outlier detection method. Our method consists of two main steps: offline modeling and online monitoring. For the offline modeling step the objectives are:To collect data $X\in {ℜ}^{m\times n}$ with m samples and n variables;To arrange each matrix X as a third-order tensor ${\underline{X}}_{I_{1},I_{2},I_{3}}$, from the one mode unfolding.

Tensors are generalizations of vectors and matrix. A zero-order tensor is a scalar, a first-order tensor is a vector, a second-order tensor is a matrix, and tensors of an order of three and higher are called high-order tensors. A tensor $\underline{X}$ of order *N* is an *N*-way array where elements $x_{i_{1},i_{2},\ldots ,i_{n}}$ are indexed by $i_{n}\in 1,2,...,I_{n}$, 1≤n≤N.

By order of a tensor we refer to the number of dimensions. The dimensions of a tensor are commonly called modes. For example, the first dimension of a tensor is mode-1, while the second is mode-2, and the third mode is mode-3, and so on. Therefore, for a three-dimensional tensor, mode-1 corresponds to the lines, mode-2 corresponds to the columns, and mode-3 corresponds to the tubes. More generally, mode-*n* corresponds to the mode-*n* fiber.

A mode-*n* fiber of a tensor $\underline{X}$ is a sub-array of elements ordered in one-dimensional form (vectors) where all dimensions are kept fixed, except for the *n*-th dimension. A mode-*n* fiber is referenced with a notation $x_{\ldots ,i_{n-1},:,i_{n+1},\ldots }$, where ":" indicates that the *n*-th dimension is varied and the others are held fixed.

Slices are two-dimensional sub-arrays (matrices) defined by fixation of all dimensions with the exception of two.

In some occasions, it is convenient to represent a tensor by a matrix. In tensor decompositions, we constantly encounter treatment of tensors in their matrixed form as a way to simplify processes that could be extremely long and confusing if they were described in the original form of a tensor. However, for this, it is necessary to define operations for manipulating tensors in this simpler form, the matrices. The first of these operations is the unfolding (or matricization).

For an *N*-dimensional tensor $\underline{X}$, there are *N* standard ways of arranging it as a matrix. Each unfolding is called mode-*n* unfolding and is denoted by ${\underline{X}}_{n}$. The mode-*n* matricization is arranging each mode-*n* fiber as columns of a new matrix, such that the order of this fibers should follow the order of the dimensions of the tensor, such that a minor dimension has a higher priority in the ordination than another superior dimension.

One of the extremely important products for tensor decomposition is the Kronecker product [21]. The Kronecker product of two matrices $A\in R^{I_{1}\times I_{2}}$ and $B\in R^{J_{1}\times J_{2}}$ is a matrix denoted by $A⊗B\in R^{I_{1}J_{1}\times I_{2}J_{2}}$ and defined as
$$A⊗B:=\left[\begin{array}{cccc} a_{11}B & a_{12}B & \cdots & a_{1I_{2}}B\\ a_{21}B & a_{22}B & \cdots & a_{2I_{2}}B\\ \vdots & \vdots & \ddots & \vdots \\ a_{I_{1}1}B & a_{I_{1}2}B & \cdots & a_{I_{1}I_{2}}B \end{array}\right]$$

One type of product that we encounter along this paper is the n-mode product between tensor and matrix. For a tensor $\underline{X}\in R^{I_{1}\times \cdots \times I_{n}\times \cdots \times I_{N}}$ and a matrix $A\in R^{J\times I_{n}}$ we denote by $\underline{X}$×${}_{n}U$ the n-mode product between those whose mathematical formulation is given by the following equation:(1)$$\begin{array}{c} {\left(\underline{X}\times_{n}U\right)}_{i_{1},...,i_{n-1},j,i_{n+1},...,i_{N}}:=\sum_{i_{n}=1}^{N}x_{i_{1},...,i_{n},...,i_{N}}u_{j,i_{n}} \end{array}$$

The equation above is the definition of the n-mode product of a tensor with a matrix via summation. Another simpler way to define such a product is to express it in terms of the matrix product between the matricization of tensor $\underline{X}$ and its own matrix U:(2)$$\begin{array}{c} \underline{Y}=\underline{X}\times_{n}U\Leftrightarrow Y_{n}=AX_{n} \end{array}$$

Higher-order tensors can be compressed through tensor decompositions if they admit a low-rank tensor approximation; this principle facilitates big data analysis [22]. The idea of *n*-rank was introduced by Kruskal [23] in 1988. Kruskal proved that, under certain explicit conditions, the expression of a third-order tensor (i.e., a three-way array) of rank $i_{r}$ as a sum of $i_{r}$ tensors of rank 1 is unique. Thus, the CANDECOMP/PARAFAC (CP) decomposition factorizes a tensor into a sum of rank-one tensors [20]. It is a special case of Higher-Order Singular Value Decomposition (HOSVD) when its core tensor is superdiagonal [20]. For example, tensor data $\underline{X}$∈${ℜ}^{I_{1}\times I_{2}\times I_{3}}$ can be decomposed as follows:(3)$$\begin{array}{cc} \underline{X} & =\sum_{r=1}^{R}u_{r}^{\left(1\right)}\circ u_{r}^{\left(2\right)}\circ u_{r}^{\left(3\right)}, \end{array}$$where *R* is the rank of the tensor, i.e., the minimum number of third-order rank-one tensors that are needed to reconstruct $\underline{X}$ exactly [24]. The vector $u^{\left(n\right)}\in {ℜ}^{I_{n}}$ denotes the *r*-th column of the factor matrix $U^{\left(n\right)}=\left[u_{1}^{\left(n\right)},...,u_{R}^{\left(n\right)}\right]\in {ℜ}^{I_{n}\times R}$ along the *n*-th mode or dimension (*n* = 1, 2, 3).

However, the most general form of the above equation in which the nucleus tensor is not super diagonal is the tensor decomposition model, called HOSVD:(4)$$\begin{array}{cc} \underline{X} & =\underline{G}\times U_{1}^{\left(1\right)}\times U_{2}^{\left(2\right)}\times U_{3}^{\left(3\right)}+\underline{E}, \end{array}$$where $\underline{X}\in R^{I_{1}\times I_{2}\times I_{3}}$, $\underline{G}\in R^{P\times Q\times R}$, $U_{1}\in R^{I_{1}\times P}$, $U_{2}\in R^{I_{2}\times Q}$, $U_{3}\in R^{I_{3}\times R}$ and $\underline{E}\in R^{I_{1}\times I_{2}\times I_{3}}$. The tensor $\underline{G}$ is called the *core tensor* and its entries give the level of the interaction between the different components.

Tensor $\underline{X}$ can also be written in three different ways using the unfolding concept. In terms of the factor matrices, the unfolding matrices of $\underline{X}$, represented as ${\underline{X}}_{1}\in R^{I_{1}\times I_{2}I_{3}}$, ${\underline{X}}_{2}\in R^{I_{2}\times I_{1}I_{3}}$ and ${\underline{X}}_{3}\in R^{I_{3}\times I_{1}I_{2}}$, admit the following factorizations from the Kronecker product
(5)$$\begin{array}{cc} {\underline{X}}_{1} & =U^{\left(1\right)}{\underline{G}}_{1}{\left(U_{3}⊗U_{2}\right)}^{T}+{\underline{E}}_{1}, \end{array}$$
(6)$$\begin{array}{cc} {\underline{X}}_{2} & =U^{\left(2\right)}{\underline{G}}_{2}{\left(U_{3}⊗U_{1}\right)}^{T}+{\underline{E}}_{2}, \end{array}$$
(7)$$\begin{array}{cc} {\underline{X}}_{3} & =U^{\left(3\right)}{\underline{G}}_{3}{\left(U_{2}⊗U_{1}\right)}^{T}+{\underline{E}}_{3}, \end{array}$$where the operator ⊗ denotes the Kronecker product.

For the online monitoring step, the objectives are:To obtain the matrices factors of the HOSVD tensorial model in an iterative way through a sliding window;From the tensor decomposition model, to calculate the monitoring statistic along the sliding window;To decide whether an observation is an outlier or normal data using the maximum Mahalanobis distance threshold.

### 2.1. Offline Modeling Step

In the offline modeling step, we model the outlier detection problem as a system. The following scheme summarizes our proposed method in a diagram based on that presented by Aquino et al. [25] and also used by Souza et al. [18] in the modeling of the offline detection. However, for an online monitoring model, we adapt the modeling using a sliding window that allows us to employ fixed-length sliding windows with well-defined time intervals.
$$N|E{\left[r\right]}^{P}V{\left[r\right]}^{S(h,k)}V^{\prime }{\left[r\right]}^{\Psi }V^{\prime \prime },\Phi$$

Thus, the meaning of each notation in this diagram is presented in Table 1. An example of this model is a city (N), with our attention restricted to a critical area *E* where the application reports online the occurrence of anomalous events. The phenomenon of interest could be eight-tuple (temperature, humidity, brightness, noise, pressure, particulate matter (PM 1), particulate matter (PM 10) and particulate matter (PM 2.5)), with infinite precision in space, time and measures.

The data collection used in this study was obtained through an environmental monitoring platform called Smart Citizen [19,26,27]. The data collection, processing and modeling procedure in this article was performed similarly to that done by Souza et al. [18]. In our proposal, we adapted the method proposed by Souza et al. [18] of the sampling $\Psi_{i}$ of three functions, to a new method composed of two functions since we did not use the clustering analysis function. Therefore, sampling was composed of the functions a HOSVD reduction of dimensionality ($\Psi_{H}$) and online outlier detection function ($\Psi_{O}$) through the sliding window:$$\Psi_{i}=\psi_{H}\circ \psi_{O},$$

To model the data to use the multidimensional HOSVD method ($\psi_{H}$), we organized the set of all multivariate observations $V^{\prime }$ in a third-order tensor ${\underline{X}}_{I_{1},I_{2},I_{3}}$, where $I_{1}$ corresponds to the time dimension, $I_{2}$ the sensed variables, and $I_{3}$ the cities analyzed. Therefore, each matrix $V^{\prime }$ (in total, three multivariate observations series representing a respective city) is considered as a slice of tensor $\underline{X}$. Therefore, through a sliding window, an observation model of the most recent flow data is generated, as presented in the following section.

### 2.2. Online Monitoring Step

In the online monitoring step, the projection distance of each data vector decomposed by the multidimensional method in the subset defined by each selected main component is calculated as new data arrives and an outlier statistical index was calculated.

Thus, to identify outliers we applied the function $\psi_{H}$ to reduce dimensionality and discover possible associations between the components of the multiway tensor. After dimensionality reduction, the function $\Psi_{O}$ was applied for outlier detection. Within this function we embedded the sliding window method to capture the continuous changes of the similarity statistical characteristics in a timely and rapid manner. The central idea was to obtain from the composition of the two functions ($\Psi_{i}$) an improvement in the detection accuracy in online monitoring. Then, as the window moves the entire process of multidimensional data decomposition is performed for each sample unit considered in the time series under analysis, the Mahalanobis distance is calculated, and a threshold is generated for each detection result.

Mahalanobis distance is used for its advantage of being affine invariant, while other methods are invariant only under certain orthogonal transformations [28]. Moreover, it should be noted that the classical covariance matrix used in the calculation of Mahalanobis distance is centered on the arithmetic mean vector, which minimizes data variation and is therefore an informative measure which considers the arithmetic mean as the data center.

Moreover, we highlight that, similarly to work by Souza et al. [18], the components of the model were selected using the criterion based on the explained variance of each component. The number of principal components of each factor matrix were chosen based on the cumulative percentage of variance explained [29]. Therefore, if the cumulative percentage of the first components is above a threshold (for example, 75% [30]), the appropriate number of components is selected as the components that exceed this limit.

### 2.3. Outlier Definition and Detection

In the outlier detection function $\psi_{O}$, we calculated the projection distance of each group vector in the subspace defined by the selected component with higher variance. For this, the distance used was that of Mahalanobis, also known as Hotelling’s $T^{2}$ statistic, a common metric for monitoring time series, which is computed as follows [31]:(8)$$\begin{array}{cc} T_{t}^{2} & ={\left(x_{t}-\bar{x}\right)}^{T}S^{-1}{\left(x_{t}-\bar{x}\right)}^{T}, \end{array}$$where $\bar{x}$ is the mean, $x_{t}$ is the multivariate observation at time *t* and S is the covariance matrix.

Based on the result of this metric, we classified an observation at instant $t_{i}$ over $V_{t_{i}}^{\prime }$ as a normal condition, if the calculated value for Mahalanobis distance ($T_{t}^{2}$) was below the Mahalanobis distance control limit ($T_{\alpha }$), that is, $T_{t}^{2}<T_{\alpha }$, conversely, we classified it as an outlier, if the Mahalanobis distance ($T_{t}^{2}$) was equal to or exceeded the Mahalanobis distance control limit ($T_{\alpha }$), that is, $T_{t}^{2}\geq T_{\alpha }$. The approximate limits of Mahalanobis distance control, with a confidence level α, can be determined in different ways by applying the probability distribution assumptions [32]:(9)$$\begin{array}{cc} T_{\alpha } & =\frac{d(n^{2}-1)}{n(n-d)}F_{\alpha }\left(d,n-d\right), \end{array}$$where $F_{\alpha }\left(d,n-d\right)$ is the upper limit of the percentile of the *F* distribution with degrees of freedom *d* and n−d. Thus, if $T_{t}^{2}>T_{\alpha }$, that is, greater than the upper limit, then the observations are considered outliers, otherwise normal:$$\begin{array}{c} \left\{\begin{array}{cc} T_{t}^{2}\geq T_{\alpha }\to & \Phi =\Phi \cup V_{t_{i}}^{\prime }\\ T_{t}^{2}<T_{\alpha }\to & V^{\prime \prime }=V^{\prime \prime }\cup V_{t_{i}}^{\prime } \end{array}\right. \end{array}$$where Φ is all outliers detected and $V^{\prime \prime }$ is the free outliers data, as depicted in the diagram above.

#### Sliding Windows in Outlier Detection

Our online outlier detection approach is based on sliding windows, that is, at each time point, it checks back a constant amount of time, referred to as a window [33]. Since the stream is continuously updated with fresh data, it is impossible to maintain all of them in the main memory. Therefore, a window is used which keeps track of the most recent data and all decomposition and detection tasks are performed based on what is “visible” through the window. That is, we use sliding windows to restrict our attention to recent data, because the time series are noisy and may change their behaviors over time, i.e., they are nonstationary. In this context, old data can add bias to the inference on recent data. Thus, the application of the sliding window technique is helpful for tracking the time-variant dynamics of the process in data, not only dealing with nonstationarity, but also reducing the computational cost of the algorithm and storage requirements, so that they are suitable for online detection. Therefore, for the data within the window, we performed the detection statistics.

The tensor ${\underline{X}}_{I_{1}I_{2}I_{3}}$ is periodically sampled at the time points along dimension $I_{1}$, for the sensed variables in the analyzed cities. Thus, a multidimensional flow is a stream of data lines of tensor $\underline{X}$ that encompasses the three dimensions ($I_{1}$, $I_{2}$, $I_{3}$), that is, by setting the dimension $I_{1}$ and varying the dimensions $I_{2}$ and $I_{3}$ we have the current sample unit. Figure 1 shows the scheme of the proposed method, where when we fix the dimension $I_{1}$, for example t1 (see dashed line in Figure 1), we have the first instant of the time series along the variables (dimension $I_{2}$) in the respective cities (dimension $I_{3}$). Our temporal window has a length of 24 h, in which after the method is applied to each sample unit of the data tensor after 24 h we pass to the second time window in which we discard the first element (instant t1) of window 1 and consider the time (t25) for window 2 (Figure 1). In addition, within each window, the Mahalanobis distance is calculated on each sample unit returned by the multidimensional decomposition as the window moves. The complete online monitoring procedure is presented through the following Algorithm 1.

**Algorithm 1:** Outlier detection algorithm.  **Data**: X_I1I2I3 - decomposed data;   **Result**: Outliers;1 Select features and save on Nf; 2 Select the size of the moving window and save on wt; 3 Select the time shift within window and save on st; 4 numit← floor (X_I1−wt)/st; 5 for i = 1:1:numit6 t0 = 1 + (i-1)st; 7 HOSVD(X_(t0:t0+wt-1, :, :),Nf); 8 Tt2← compute; 9 Tα← compute; 10 **return** Outliers. 

The thresholds used for outlier detection are based on the results found from the Mahalanobis distance calculation. Based on the result of this metric, we classify an observation as a normal condition, if the calculated value for Mahalanobis distance is below of Mahalanobis distance control limit, or we classify the observation as an outlier, if the Mahalanobis distance is equal to or exceeds the Mahalanobis distance control limit.

## 3. Results and Discussion

In this section, we illustrate our online outlier detection method as presented with the aim of detecting outliers of the monitored environmental variables of cities urban spaces. A comparison of individual performance was performed on the basis of simulations and the results are compared with the results obtained by Souza et al. [18].

### 3.1. Real Data

The boxplots of the real values collected by the sensors of the platform Smart Citizen of the cities of Helsinki, Tuusula, and Lohja are presented in Figure 2 and Figure 3. We considered outdoor sensors for a period of 16 days (December 1 through December 16, 2018), totaling a bank of 381 h of monitoring of the eight environmental variables. These locations were selected because they offer online sensor nodes where measurements can be performed in real time without missing data.

Observing the data behavior pattern, we noticed that the brightness and particulate matter variables (PM 1, PM 10, and PM 2.5) were the ones that presented more discrepant values in relation to the other variables. This behavior of the variable brightness is explained by the alternation between peaks and valleys of their values since, at dusk, the luminosity in the cities is reduced considerably. On the other hand, for the variables PM 1, PM 10, and PM 2.5, the discrepant behavior may be related to the area with significant atmospheric pollution.

### 3.2. Online Detection

The idea behind the sliding window is to process the data in smaller batches at a time, usually to represent a neighborhood of points in the data. Therefore, using a fixed band of 24 h we updated the data every hour, that is, as new data arrived at a given instant, the method was updated. The choice of updating the sliding window every hour over a fixed 24 hour window was decided, as it was determined that with an increase in this period, the shorter peaks can be eliminated and the outliers can be camouflaged. That is, if the window is too large, the window may contain outdated information, and the accuracy of the model decreases [33]. Thus, from the results of the tensor decomposition, we analyzed the temporal dimension once we focused on the analysis of the time series of the model. Figure 4 presents online monitoring models in which every hour, that is, with each moment in which new data arrive, the decomposition of the multidimensional model is updated. Thus, our sliding window moves along a fixed window of 24 h, and throughout this window, the data are updated with a granularity of 1 hour until the entire time series is contemplated, where we observe throughout the process the dynamics of the temporal behavior of the data and its effects on the rest of the sliding window band as it moves. For example, Figure 4 presents online monitoring for the first and second day and the change in the dynamics of temporal behavior as new data are incorporated into the multidimensional model. For a better understanding, the first subplot of Figure 4 shows the first 24 h considered in the model analysis, then the window moves (moving to the second day) every 1 hour, and as a new die enters, the last die of the time series is discarded, and so the model is updated. This process is repeated until the entire time series is contemplated.

For a better visualization, Figure 5 shows the first subplot of Figure 4, where we observed online monitoring (with the sliding window) and added a comparison with offline monitoring for the first 24 h monitored, in addition to establishing detection limits for each. We observed that online monitoring through the sliding window (red line) identifies a greater number of peaks, pointing to a greater variation in data dynamics than in relation to offline monitoring that identifies only a single more expressive valley with its respective two peaks (blue line). As valley points are surrounded by two larger neighbors (immediately anterior and posterior), this result corroborates the hypothesis that offline monitoring does not represent a good approximation of data for monitoring, unlike online monitoring that represents a better approximation of these points revealing the granularity of the outliers. This is because unlike traditional data mining, which can read time series static over and over again, each sample in a data stream is examined along the sliding window [33].

Another important change is in the analysis of the variation of the limit that is updated according to the progress of the control window of its value, that is, from the perspective of an online monitoring window generating a new threshold. On the other hand, from the perspective of offline monitoring, this limit was not included in the whole dataset. Thus, Figure 6 shows the threshold dynamics of identification of the outliers, in which fluctuations are observed throughout the time series. The analysis of a dynamic threshold in outlier detection from the perspective of urban environmental monitoring is still scarce in the literature. However detecting outliers from the perspective of network traffic has been widely studied. For example, it is observed in some works such as [34,35] that dynamic thresholds improve outlier detection accuracy, while static thresholds result in a lower detection accuracy. In the context of urban environmental monitoring this fact can be verified when comparing the dynamic threshold used in this work with the static threshold used in the work of Souza et al. [18], which presented a lower precision.

As the sliding window for a stream moves and old data expire and new data come in, it is possible to discover the outliers for data streams at any time instant. In this perspective, Figure 7 presents the variables that were along all the windows (discriminated on the x-axis), from the input data stream in the model, from the moment that a window that presents/displays a greater amount of discrepant values is presented, and which is the window that receives the smaller number of outliers. In addition, we can also consider a range of sliding windows and observe the dynamics of the arrangement of these outliers over the interval according to temporal evolution. Consider the window w2 as the one that presented the largest number of outliers, in which we observed a total of nine outliers, while the window w12 was the one that presented the lowest number of discrepant values with only one outlier. We found that the intervals between windows w1 and w10 were those with a higher concentration of outliers, while the remaining windows had the number of outliers falling (mainly in windows w11 and w12), going from an average of 4.5 outliers per window to an average of 3.5. As a whole, the monitoring results in 16 windows, totaling 66 outliers. In addition, the pattern of events generated changes with the sliding data window, thus, it is a variable pattern detection model. This method can capture the dynamics of a time-varying system, and it is suitable for describing data behavior from time-varying urban environmental monitoring.

### 3.3. Performance Evaluation

In this subsection, we discuss the methods used to evaluate the performance of online outliers detection. We evaluated the performance of multidimensional approaches, HOSVD online and MPCA (Multilinear Principal Component Analysis) online methods concerning the receiver operational characteristic (ROC). In addition to that, an approach to evaluate the statistical significance of an ROC curve is to calculate the area under the curve (AUC). Thus, the results shown in Figure 8 exhibit an even higher AUC (0.80 in the picture) for HOSVD online and a lower, AUC (0.65 in the picture) for MPCA online. The curves also reveal that because the AUC for online MPCA is smaller than the AUC of HOSVD online, this phenomenon corroborates the results found for the online HOSVD method that detected a larger number of outliers. Although the gain found for the online HOSVD method of 0.80 accuracy approached the accuracy of the method proposed by Souza et al. [18], 0.87, this result points out that our online method can achieve more significant gains once it is combined with clustering algorithms such as k-means. Therefore, we observed that the HOSVD tensor decomposition method combined with the sliding window detection in online monitoring is established with greater precision. This phenomenon can be gauged by the fact that in this configuration, the data structure is richer in information than when the data are unfolding for the application of the MPCA method.

In addition, we performed a performance comparison using the same data used in Souza et al. [18], i.e., for the cities of Elda, Rois, Tallinn, and Nuremberg over 15 days from July 1st, 2017 to July 15th, 2017. The results found in [18] revealed that the HOSVD+kmeans method found about 71 outliers for the first main component selected (41 outliers for cluster I and 30 outliers for cluster II), while for the second main component selected 62 outliers (30 outliers for cluster I and 32 outliers for cluster II) were detected. For the MPCA+kmeans method, 35 outliers for the first main component were detected (20 for cluster I and 15 for cluster II), while for the second main component 22 outliers were detected (10 outliers for cluster I and 12 outliers for cluster II). When we applied both the online HOSVD method and the online MPCA method for the same data used in the article [18], we found that our online method through the sliding window detected 71 outliers (for the HOSVD online method) and 35 outliers for the MPCA method online). These results, when compared with the results of [18], were found to correspond to the outliers detected by the first components for both clusters found (Cluster I and Cluster II) of both HOSVD+kmeans and MPCA+kmeans. That is, online methods HOSVD and MPCA detected 41.08% and 20.21%, respectively. This experiment shows that for the same data used in Souza et al. [18], the tensor decompositions HOSVD and MPCA combined with the sliding window identified outliers similarly to the HOSVD+kmeans and MPCA+kmeans methods. That is, the sliding window results in our online method showing better computational performance when compared to the offline method, in addition to saving memory space [30].

Again, we performed a performance comparison using the ROC curves, this time between the proposed online methods and the offline methods used in [18]. For this, we compared the outlier detection performance in both clusters I and II for both offline HOSVD+kmeans and MPCA+kmeans methods compared to HOSVD and MPCA online detection methods. Figure 9 shows the ROC curves, where a greater accuracy was found for the online HOSVD method compared to the HOSVD + kmeans method, with an accuracy of 0.98. The same pattern of superiority of the MPCA online method was observed when compared to the offline MPCA+kmeans method, with an accuracy of 0.73 (Figure 10). Therefore, we observed that the online methods presented supremacy over the offline methods, both in Figure 9 and Figure 10. Moreover, this phenomenon of the superiority of the online HOSVD method over the MPCA method is due to the fact that the HOSVD model presents a richer three-dimensional information structure, since the MPCA method has its data structure matrixed in mode 1. That is, the decomposition of the MPCA method occurs over a two-dimensional structure, whereas the HOSVD decomposes the data considering the three analyzed dimensions (time, measurements and space).

Thus, the results showed greater efficiency for the online method of detection of outliers when compared to the offline approach in terms of accuracy. When averaging the offline approaches to the HOSVD method (Figure 9), we have an average accuracy of 88%, whereas for the online approach, we have an accuracy of 98%, with a gain of 10% in the accuracy of detection online. On the other hand, when averaging the offline approaches for the MPCA method (Figure 10), we have an average accuracy of 64.5%, while for the online approach we have an accuracy of 73%, with a gain of 8.5% in the accuracy of online detection. Therefore, the gain in terms of accuracy of online detection ranges from 8.5% to 10% gain.

## 4. Conclusions

In this paper, we proposed an online method for outlier detection in environmental data from the monitoring of smart cities and to deal with online tensor data based on multiway decomposition. The proposed HOSVD online method, which uses a sliding window to provide online detection, aims to extract the slowly varying features that are the representations of the occurrence instants of a particular event by efficiently extracting the process dynamics. Moreover, we contributed to the literature by incorporating a new incremental tensor analysis, known as ITA [36]. Our contribution focuses on the window-based tensor analysis, where instead of processing individual tensors we used a sliding window strategy to handle time dependency between consecutive tensors [30].

Contrary to MPCA online, HOSVD online detects outliers with greater accuracy. The simulation study together with the data analysis illustrates that HOSVD online consistently detects outliers, when they are present, with a small proportion of false detections, while the success of its competitors depends more on the data set under study. While online MPCA performs better in terms of accuracy compared to MPCA offline (MPCA+kmeans, [18]), HOSVD online continues to perform even better on HOSVD offline (HOSVD+kmeans, [18]). This result gives us a new, more efficient approach to the big data analysis of urban environments in smart cities from online monitoring and detection of outliers.

In addition to that, we conclude that online detection of outliers from real-time monitoring through the sliding window becomes a more adequate approach when compared to offline detection (as compared to [18]), since in the offline approach high memory resources and a higher processing load are required as the window has a high fixed width. This result is corroborated through the accuracy of both approaches, with superiority in the online approach. In our future research, we plan to continue exploring our proposed approach in the following three aspects: first, to incorporate the sliding window in the other modes of the multidimensional decomposition, considering the dynamic aspect of the other dimensions, as well as to explore other kinds of window models, such as landmark window, tilted window and fading window [33]. Second, to propose new approaches online using other multidimensional decompositions. Third, we will look for many other real-life applications of our proposed approach, such as false data injection detection in smart cities, in addition to evaluating the reduction of false alarms.

## Figures and Tables

**Figure 1 sensors-19-04464-f001:**
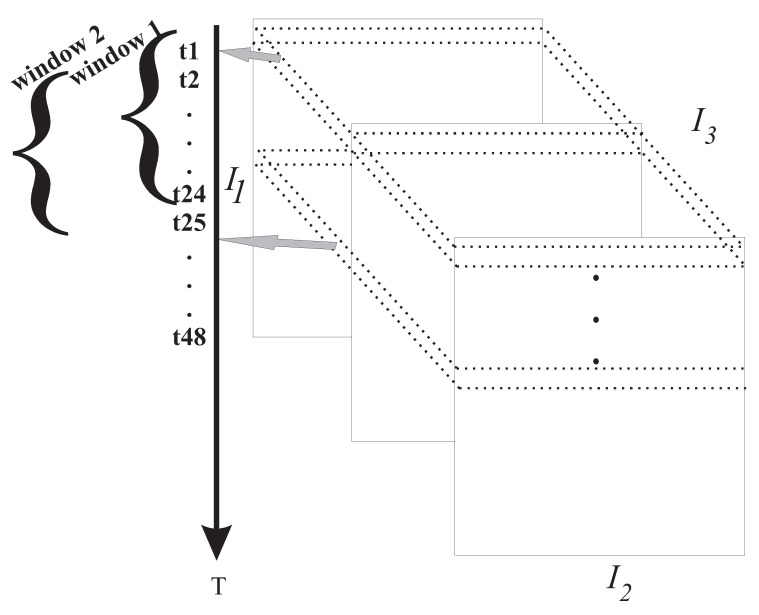
Monitoring online: Sliding windows.

**Figure 2 sensors-19-04464-f002:**
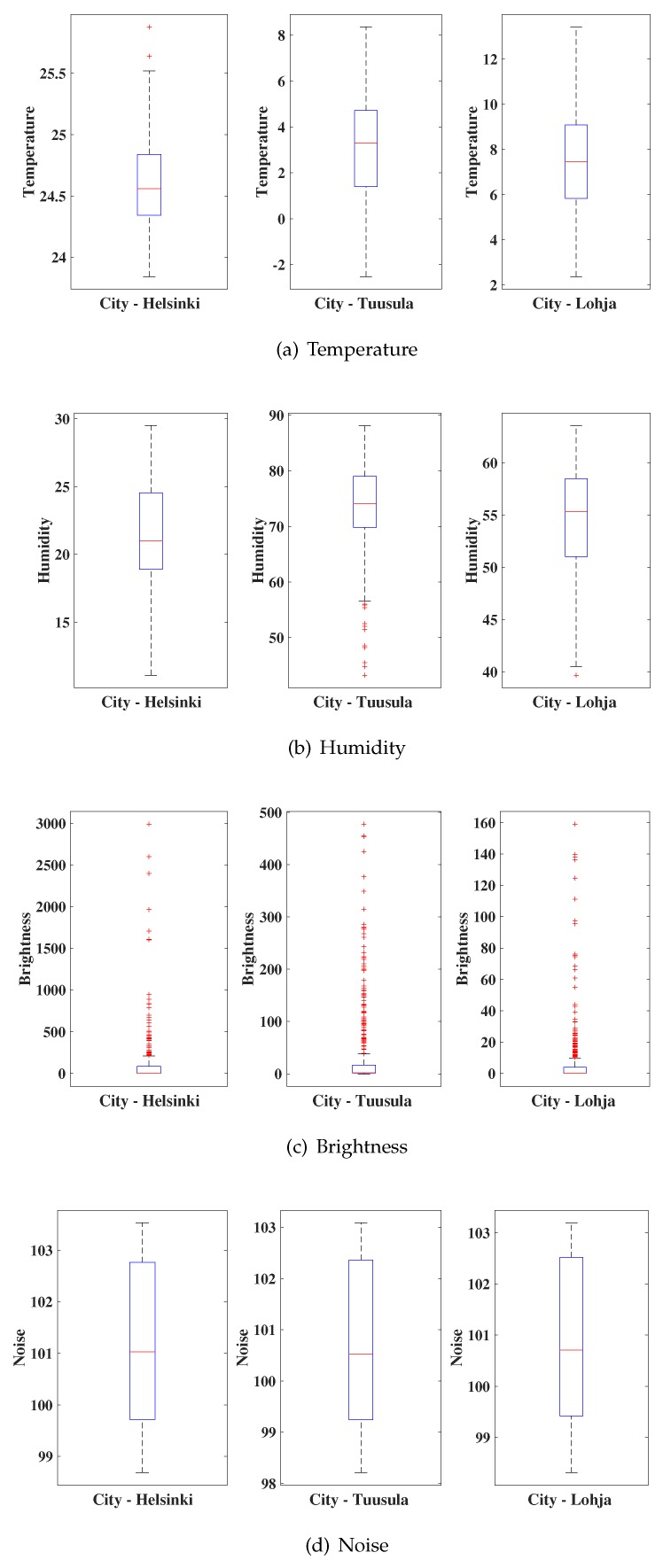
Time series of data collected.

**Figure 3 sensors-19-04464-f003:**
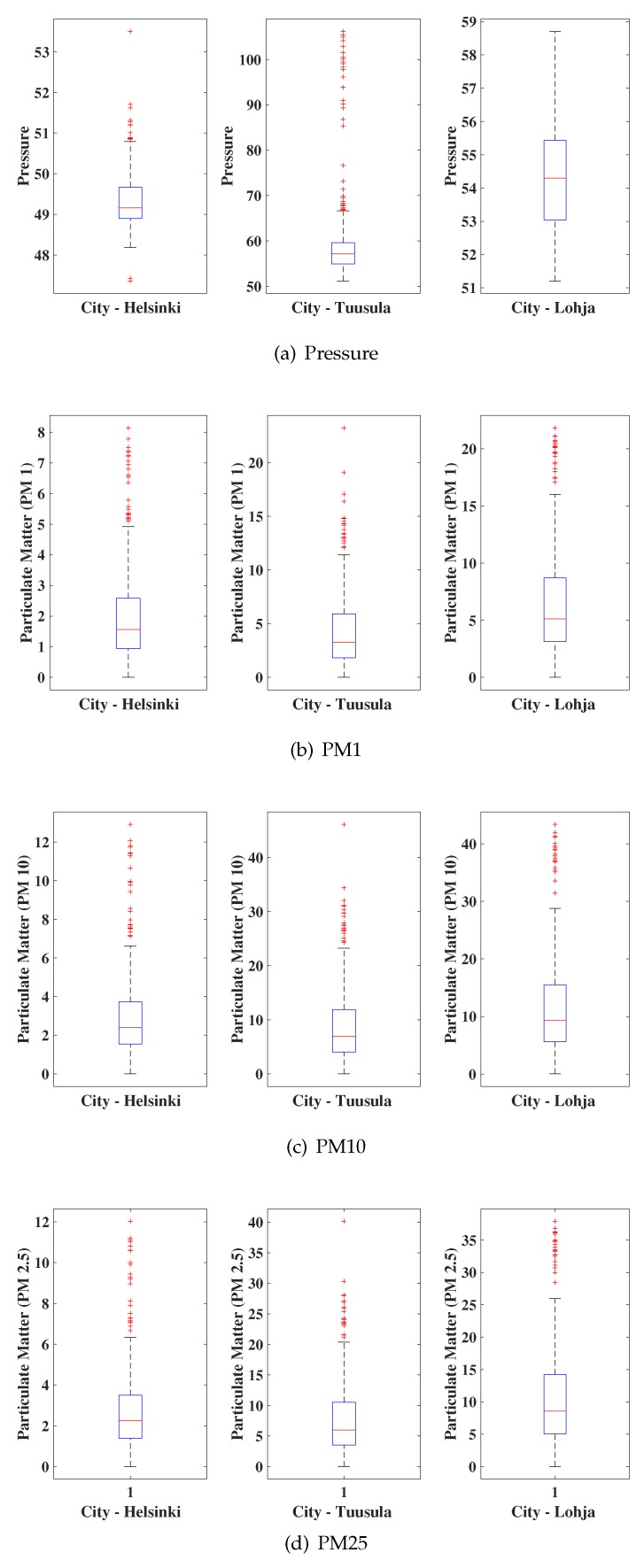
Time series of other data collected.

**Figure 4 sensors-19-04464-f004:**
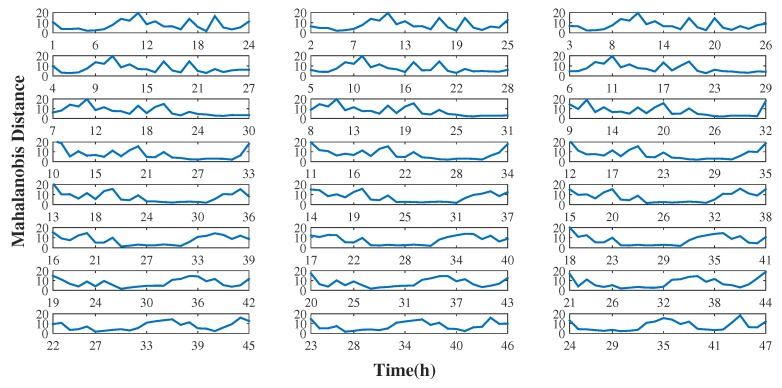
Online monitoring - Day #1 and Day #2.

**Figure 5 sensors-19-04464-f005:**
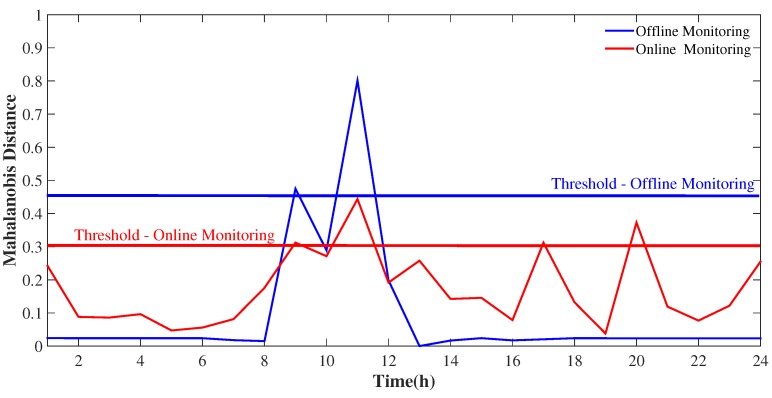
Online monitoring versus offline monitoring - Day #1.

**Figure 6 sensors-19-04464-f006:**
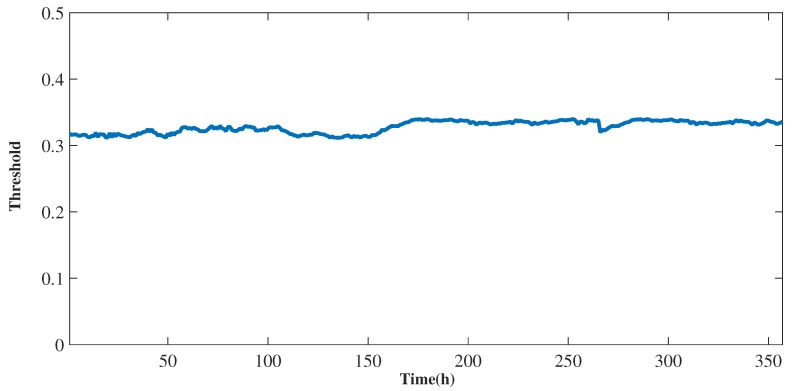
Threshold of online monitoring.

**Figure 7 sensors-19-04464-f007:**
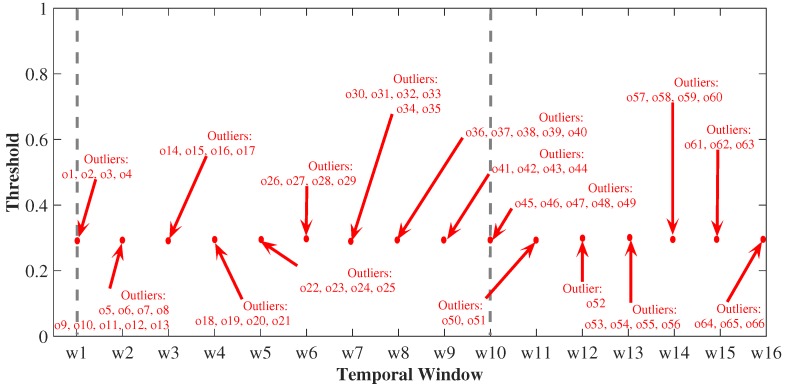
Outliers detection for sliding windows.

**Figure 8 sensors-19-04464-f008:**
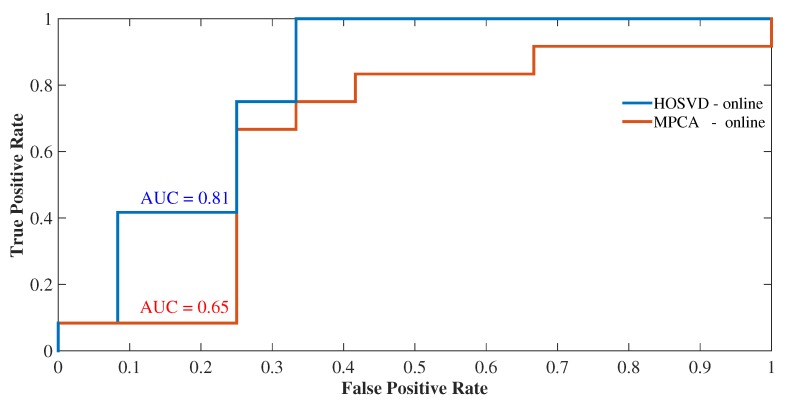
Detection performances of the methods, HOSVD online and MPCA online.

**Figure 9 sensors-19-04464-f009:**
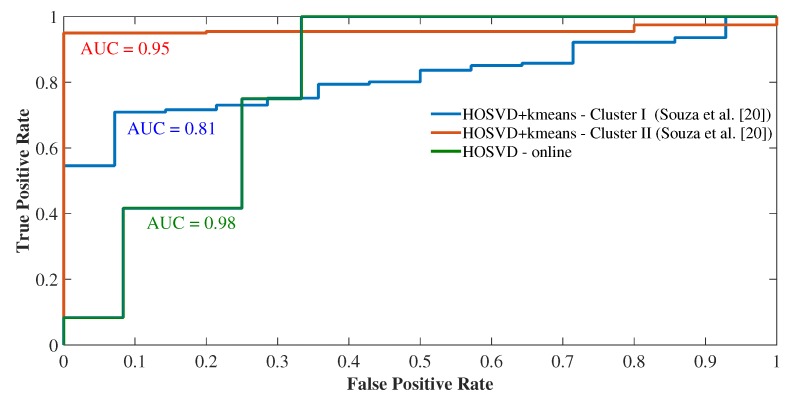
Detection performances of the methods HOSVD offline × HOSVD online.

**Figure 10 sensors-19-04464-f010:**
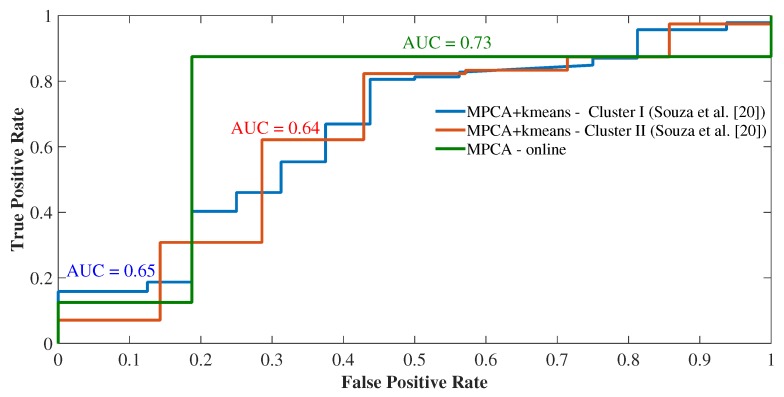
Detection performances of the methods MPCA offline × MPCA online.

**Table 1 sensors-19-04464-t001:** Diagram meaning.

Notation	Meaning
N	represent the environment and the process to be measured
∣	the study restricted to *E*
*E*	time-space domain and topological characteristics of the monitored area
*P*	phenomenon of interest
V	represent the domain, i.e., is the set of all possible phenomena
S	$S=(S_{1},\ldots ,S_{o})$ set of *o* observer nodes
*h*	denotes the collection of all positions of each node
*k*	denotes the set of all characteristic functions
$V^{\prime }$	a real-valued vector
Ψ	is the set of all operations on each node, 1≤i≤o: $\Psi =(\Psi_{1},\cdots ,\Psi_{o})$.
$V^{\prime \prime }$	$V^{\prime \prime }$ is the free outliers data
Φ	is all outliers detected

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
