# Peer review of "An Online Method to Detect Urban Computing Outliers via Higher-Order Singular Value Decomposition"

_sensors, 2019, doi:10.3390/s19204464_

Round 1
Reviewer 1 Report
This manuscript presents an online method to explore the multiway nature of urban spaces in outliers inspection. This method considers two stages: offline modeling stage and online modeling stage. However, this manuscript must be improved considering the following comments:
1.-A papers should contain four sections: introduction, methods and materials, results and discussion, and conclusions. This manuscript has two sections with few information: related work and background. The sections of the manuscript should only consider four sections (i.e., introduction, methods and materials, results and discussion, and conclusions)
2.-Authors must improve the English style of the manuscript. This manuscript contains many large sentences. The sentences should have between 17 and 25 words. For instance, the first sentence of the introduction has 53 words:
The integration between Information and Communications Technology (ICT), cloud computing, and Internet of Things (IoT) in smart cities favors the consolidation of an urban environment which integrates multiple information and communication technology in the management of the various services offered by the city, such as: transportation systems, education, health, safety, and environmental monitoring.
In addition, section of related work has the following two large sentences:
In energy field, Wang et al. [21] apply in a large dataset of monthly energy use data of residential buildings with highly-granular, three different methods including the standard deviation method, quartile method, and Grubbs’ test for outlier recognition to improve the performance of reverse modeling techniques, identifying not only the outliers but also the causes that generated such outliers, that is, the types of household appliances that increased the energy consumption of the residences.
The work [22] proposes in an industrial perspective, an outlier detection scheme that can be directly used for either industrial process monitoring or process control from developing several detection algorithms, of which the mean function, covariance function, likelihood function and inference method are specially devised based on traditional Gaussian process regression.
3.-Abstract section. The first sentence of this section should include an introduction. This section should add more results and a sentence of conclusion.
4.-Introduction section. This section must contain the main advantages and challenges of the proposed method with respect to other methods reported in the literature.
5.- Between lines 258 and 262, the following sentences must include references.
The number of principal components of each factor matrix is chosen based on the cumulative percentage of variance explained [? ]. Therefore, if the cumulative percentage of the first components is above a threshold (for example, 75% [? ]), the appropriate number of components is selected as the components that exceed this limit. Thus, this step presents a way of drastically reducing the dimensionality of the dataset.
6.- In Figure 2, the size of the words (window 1 and 2, t1,...t48) must be reduced. The size of the labels and symbols of the Figures 3(a-d) and 4(a-d) must be increased. Size of the Figure 4 must be increased.
7.- Authors must include more critical discussion about the results shown in Figures 5, 6, 7, 8, 9, 10 and 11. In addition, authors must consider the main disadvantages or challenges of the proposed method.
Reviewer 2 Report
The paper contains an interesting proposal for online monitoring of urban spaces using collected data and taking into account offline modeling, which is important for online implementation validation. The online detection is very acceptable. The structure of the work is well planned. However, some things must be clarified:
What thresholds of detection of outliers values were used?
- How did they evaluate the decrease of false alarms in the detection? Are the data affected by noise? If the data are affected by noise, how these could affect the detection method.
Since they only present the performance evaluation, but not the reduction of false alarms.
Reviewer 3 Report
The paper describes an interesting study and generally flows well.
The paper needs some clarification, justification, reorganisation and summarisation to improve it.
Comments ordered by line number:
There is no figure 1 - I assume the cross-references are valid except the reference to figure 1.
Abstract line 7-9. The final sentence is not clear. What is online and offline here? They have not been described. It would be good to clarify the difference as the terms are used throughout the manuscript.
Line 22, a general outlier reference would be useful here (e.g., Hodge V, Austin J. A survey of outlier detection methodologies. Artificial intelligence review. 2004 22(2):85-126)
Line 24, the reference seems incorrect, the sentence is general but the reference very specific. Could you use Chandola V, Banerjee A, Kumar V. Anomaly detection: A survey. ACM computing surveys (CSUR). 2009 41(3):15
Line 53 would benefit from a brief description of article [13] as the reader may not have access to article [13] while reading this manuscript. Perhaps the description on lines 62 - 65 could be merged?
Line 58 'big' and 'data' are repeated.
Line 116 describes a sliding window approach as novel. Sliding windows have been used for years and are relatively standard. Is there an aspect of the sliding window that is distinct?
Line 151, reference for Kronecker product (e.g., H.V. Henderson, F. Pukelsheim, S.R. Searle, On the history of the Kronecker product,Linear and Multilinear Algebra, 14 (1983), pp. 113-120
What is the Kronecker product and why is it used?
Line 161 missing citation.
Line 190 why is the Mahalanobis distance used?
Section 4.2.1 lines 208- The text should give more details of the data. How often are they collected, what is the data size, what are the features, are data missing, are features missing?
Lines 259 & 260 missing citations
Line 314-318, the sentence is very long and difficult to read.
Figures 3-11 some of the text is too small to read and needs enlarging.
Round 2
Reviewer 1 Report
Authors have improved their manuscript considering all the reviewer's comments. This manuscript is suitable to be published in Sensors.
Reviewer 3 Report
The authors have addressed the changes that I suggested. I have now recommended accept.